# Simultaneous Treatment of 5-Aminosalicylic Acid and Treadmill Exercise More Effectively Improves Ulcerative Colitis in Mice

**DOI:** 10.3390/ijms25105076

**Published:** 2024-05-07

**Authors:** Jun-Jang Jin, Il-Gyu Ko, Lakkyong Hwang, Sang-Hoon Kim, Yong-Seok Jee, Hyeon Jeon, Su Bee Park, Jung Won Jeon

**Affiliations:** 1Team of Efficacy Evaluation, Orient Genia Inc., Seongnam 13201, Republic of Korea; threej09@hanmail.net (J.-J.J.); lhwangphd@gmail.com (L.H.); 2Research Support Center, School of Medicine, Keimyung University, Deagu 42601, Republic of Korea; rhdlfrb@naver.com; 3Department of Neurosurgery, Robert Wood Johnson Medical School, Rutgers, The Stat University of New Jersey, Piscataway, NJ 08854, USA; sanghoon.kim1@rutgers.edu; 4Research Institute of Sports and Industry Science, Hanseo University, Seosan 31962, Republic of Korea; jeeys@hanseo.ac.kr; 5Department of Computer Science and Engineering, College of Engineering, Seoul National University, Seoul 08826, Republic of Korea; hj@hcil.snu.ac.kr; 6Department of Internal Medicine, Kyung Hee University Hospital at Gangdong, College of Medicine, Kyung Hee University, Seoul 05278, Republic of Korea; endobee90@gmail.com

**Keywords:** ulcerative colitis, treadmill exercise, combined treatment, inflammation, apoptosis

## Abstract

Ulcerative colitis (UC) is characterized by continuous mucosal ulceration of the colon, starting in the rectum. 5-Aminosalicylic acid (5-ASA) is the main therapy for ulcerative colitis; however, it has side effects. Physical exercise effectively increases the number of anti-inflammatory and anti-immune cells in the body. In the current study, the effects of simultaneous treatment of treadmill exercise and 5-ASA were compared with monotherapy with physical exercise or 5-ASA in UC mice. To induce the UC animal model, the mice consumed 2% dextran sulfate sodium dissolved in drinking water for 7 days. The mice in the exercise groups exercised on a treadmill for 1 h once a day for 14 days after UC induction. The 5-ASA-treated groups received 5-ASA by enema injection using a 200 μL polyethylene catheter once a day for 14 days. Simultaneous treatment improved histological damage and increased body weight, colon weight, and colon length, whereas the disease activity index score and collagen deposition were decreased. Simultaneous treatment with treadmill exercise and 5-ASA suppressed pro-inflammatory cytokines and apoptosis following UC. The benefits of this simultaneous treatment may be due to inhibition on nuclear factor-κB/mitogen-activated protein kinase signaling activation. Based on this study, simultaneous treatment of treadmill exercise and 5-ASA can be considered as a new therapy of UC.

## 1. Introduction

Inflammatory bowel disease (IBD) includes the heterogeneous forms of Crohn’s disease and ulcerative colitis (UC), both of which have alternating active and quiescent states that occur cyclically in nature [1]. UC is characterized by persistent mucosal ulceration of the colon that begins in the rectum and causes severe symptoms including abdominal pain, diarrhea, rectal bleeding, and malnutrition [2].

UC follows an unpredictable course, which can lead to complications leading to hospitalization, surgery, and escalation of treatment. The incidence of UC has increased rapidly in Asian countries, and the peak age of onset for UC is 20–30 years [3]. It is suggested that the exact mechanism of UC is likely to be related to genetic factors, immune system dysfunction, infection, and environmental factors, but the exact cause is unknown [4]. Abnormal activation of nuclear factor-κB (NF-κB) and mitogen-activated protein kinase (MAPK) signaling pathways and overexpression of pro-inflammatory cytokines such as tumor necrosis factor (TNF)-α, interleukin (IL)-1β, and IL-6 are the major factors in the onset and progression of UC [5]. Additionally, activation of the NF-κB/MAPK signaling pathway can cause multiple responses, such as apoptosis and cell proliferation, especially in UC patients [6]. 

5-Aminosalicylic acid (5-ASA) is the main drug of UC. Although its action mechanism is not fully understood, 5-ASA appears to act locally and exert anti-inflammatory effects by inhibiting local prostaglandin and leukotriene synthesis in the gastrointestinal colonic mucosa [7]. Nonetheless, inflammation relief is not permanent and does not prevent symptom recurrence. Therefore, new treatments for UC that improve colon homeostasis and effectively suppress inflammation are needed.

Physical exercise is an effective way to improve body function and increase homeostasis without causing any side effects. Regular exercise effectively increases anti-inflammatory and anti-immune cells in the body [8]. Pro-inflammatory cytokine expression was suppressed and anti-inflammatory cytokine expression was enhanced by freewheel exercise in mouse intestinal lymphocytes [9]. In a mouse model, physical exercise improved acute UC symptoms by suppressing colonic inflammation [10]. Exercise enhances anti-inflammatory and anti-immune cells and reduces apoptosis, but the protective effect of simultaneous treatment of treadmill exercise and 5-ASA against UC has not yet been identified. Here in this study, we studied the effect of simultaneous treatment of treadmill exercise and 5-ASA in mice with dextran sulfate sodium (DSS)-induced UC.

Body weight, disease activity index (DAI), colon weight, and colon length were measured. Histological analysis, hematoxylin and eosin (H&E) staining, Masson’s trichrome staining, and terminal deoxynucleotidyl transferase-mediated deoxyuridine triphosphate nick end labeling (TUNEL) assays were carried out. Western blot was conducted to identify the expression levels of TNF-α, IL-1β, IL-6, p-38, c-Jun N-terminal kinase (JNK), extracellular signal-regulated kinase (ERK), NF-κB, NF-κB inhibitor-α (IκB-α), B-cell lymphoma-2 (Bcl-2), and Bcl-2-associated X protein (Bax). Immunohistochemistry was done to measure cleaved caspase-3 and cleaved caspase-9.

## 2. Results

### 2.1. Body Weight and DAI Score

Changes in the body weight of DSS-induced UC mice are presented in Figure 1 (bottom-left panel). Induction of UC significantly reduced body weight compared to control mice starting from the 4th day of DSS administration, with the greatest decrease occurring on the 11th day of the experiment (*p* < 0.05). However, after 14 days, body weight increased significantly in the treadmill exercise group or in the 5-ASA treatment group and continued to increase until the end of the experiment (*p* < 0.05). Although there was no statistical significance, body weight increased more in the group that used treadmill exercise and 5-ASA simultaneously.

Disease severity was measured by the DAI score, which represents a composite score of weight loss, stool consistency, and rectal bleeding. Changes in DAI score in DSS-induced UC mice are presented in Figure 1 (bottom-right panel). DAI scores were higher in DSS-induced UC mice than in control mice (*p* < 0.05). But, treadmill exercise or 5-ASA significantly reduced DAI scores, indicating that treadmill exercise or 5-ASA significantly attenuated the severity of UC symptoms (*p* < 0.05). Additionally, simultaneous treatment of treadmill exercise with 5-ASA reduced DAI scores more strongly than single treatment of 5-ASA or treadmill exercise (*p* < 0.05).

### 2.2. Colon Weight and Length

Changes in colon weight and length in DSS-induced UC mice are presented in Figure 2. UC induction significantly reduced colon weight and length compared to control mice (*p* < 0.05). But, treadmill exercise or 5-ASA significantly increased colon weight and length (*p* < 0.05), indicating that treadmill exercise or 5-ASA significantly attenuated the severity of UC symptoms. Nevertheless, there was no significant difference in colon weight or length between the simultaneous treatment group and the single treatment of 5-ASA or treadmill exercise.

### 2.3. Histological Evaluation and Collagen Deposition

Histological changes in colon tissue induced by UC are presented in Figure 3. The colonic mucosa of control mice appeared normal with intact epithelium. In contrast, DSS administration induced mucosal damage, epithelial layer loss, mucosa and mucosal glands distortion, and inflammatory cell infiltration. Treadmill exercise or 5-ASA significantly improved mucosal damage due to the regenerative effect of the colon. The histological appearance of the colon tissue in the simultaneous treatment group was similar to that of the control group.

Collagen deposition in UC-induced colon tissue is shown in Figure 3. In UC-induced mice, collagen deposition in the mucosal layer was clearly increased compared to the collagen level in control mice. Treadmill exercise reduced collagen deposition in colonic tissue, and the colonic tissue after simultaneous treatment of treadmill exercise with 5-ASA was similar in appearance to that of the control group. However, 5-ASA alone did not reduce collagen deposition.

These alterations led to an elevated colonic damage score and collagen area (*p* < 0.05). In addition, simultaneous treatment of treadmill exercise with 5-ASA reduced the colonic damage score and collagen area more than single treatment of 5-ASA or treadmill exercise (*p* < 0.05).

### 2.4. Pro-Inflammatory Cytokines Expressions

The expression levels of TNF-α, IL-1β, and IL-6, types of pro-inflammatory cytokines, in UC-induced colon tissue are shown in Figure 4. The expression levels of TNF-α, IL-1β, and IL-6 were enhanced in DSS-induced UC mice (*p* < 0.05). However, the expression levels of TNF-α, IL-1β, and IL-6 were suppressed in mice in the treadmill exercise group or the 5-ASA group (*p* < 0.05). Simultaneous treatment of treadmill exercise with 5-ASA inhibited TNF-α and IL-1β expression more strongly than single treatment of treadmill exercise or 5-ASA (*p* < 0.05).

### 2.5. NF-κB/MAPK Signaling Pathway Expressions

The levels of NF-κB activation and phosphorylation levels of MAPK cascade factors in UC-induced colonic tissues are presented in Figure 5. DSS administration caused IκB-α phosphorylation, which enhanced NF-κB expression in colonic tissues (*p* < 0.05). Treadmill exercise or 5-ASA treatment suppressed IκB-α phosphorylation, as well as NF-κB expression (*p* < 0.05). Simultaneous treatment of treadmill exercise with 5-ASA more potently suppressed phosphorylation of IκB-α than single treatment of treadmill exercise or 5-ASA (*p* < 0.05). The phosphorylation levels of p38, JNK, and ERK were increased in DSS-induced UC mice (*p* < 0.05). But, p38, JNK, and ERK phosphorylation levels were attenuated by treadmill exercise or 5-ASA (*p* < 0.05). Simultaneous treatment of treadmill exercise with 5-ASA produced a more potent suppression on phosphorylation of p38 and JNK than single treatment of treadmill exercise or 5-ASA (*p* < 0.05).

### 2.6. TUNEL-Positive, Cleaved Caspase-3,-9-Positive, and Bax/Bcl-2 Ratio

The percentages of TUNEL-positive, cleaved caspase-3,-9-positive cells and expressions of Bax/Bcl-2 in UC-induced colonic tissue are presented in Figure 6. The percentages of TUNEL-positive, cleaved caspase-3,-9-positive cells increased in the DSS-induced UC mice (*p* < 0.05). However, treadmill exercise or 5-ASA suppressed the percentages of TUNEL-positive, cleaved caspase-3,-9-positive cells (*p* < 0.05). Simultaneous treatment of treadmill exercise with 5-ASA more potently suppressed the percentages of TUNEL-positive, cleaved caspase-3,-9-positive cells than single treatment of treadmill exercise or 5-ASA (*p* < 0.05). 

Induction of UC increased Bax expression (*p* < 0.05) and suppressed Bcl-2 expression (*p* < 0.05). Therefore, induction of UC by DSS administration increased the Bax to Bcl-2 ratio. However, treadmill exercise or 5-ASA treatment decreased Bax expression (*p* < 0.05) and increased Bcl-2 expression (*p* < 0.05), resulting in a reduction in the Bax to Bcl-2 ratio (*p* < 0.05). Simultaneous treatment of treadmill exercise with 5-ASA more potently suppressed Bax expression than single treatment of treadmill exercise or 5-ASA (*p* < 0.05).

## 3. Discussion

DSS-induced UC in mice exhibits symptoms similar to human UC, destroying the colonic epithelium, causing diarrhea, weight loss, hematochezia, and inflammation [5]. As a result of this experiment, when DSS was administered orally to mice for 7 days, body weight was decreased, DAI score was increased, and colon weight and length were decreased. Weight loss and increase in DAI score in mice indicate worsening UC symptoms [11]. In this study, we assessed UC symptoms using the DAI. The results of this experiment confirmed that DSS administration induced moderate to severe UC in mice.

TNF-α, IL-1β, and IL-6, classes of pro-inflammatory cytokines, were elevated in animal models of UC [12]. High serum levels of TNF-α were found in the patients of UC and Crohn’s disease [13,14]. IL-6 plays an important role in UC pathogenesis, causing neutrophil infiltration and promoting chemotaxis-induced tissue necrosis and destruction [15,16]. Additionally, IL-6 triggers the secretion of TNF-α and IL-1β and enhances adhesion proteins expression, including intercellular adhesion molecule-1 [17]. As a result, suppressing the expression of pro-inflammatory cytokines such as TNF-α, IL-1β, and IL-6 effectively treats ulcer damage in the colon. Physical activity is known as an effective adjunctive treatment for the patient of IBD [18]. A previous study using a colitis animal model has shown that exercise reduces pro-inflammatory cytokines TNF-α and IL-1β expression, increases anti-inflammatory cytokine IL-10 expression, and reduces stress-induced barrier dysfunction. Exercise has been shown to attenuate and alleviate colitis [5]. 

It has been reported that DSS treatment very potently enhanced pro-inflammatory cytokines production [12]. In our study, DSS administration also increased the production of pro-inflammatory cytokines, including TNF-α, IL-1β, and IL-6, in ulcerative colon tissue. This means that UC becomes worse due to overproduction of pro-inflammatory cytokines. However, treadmill exercise and 5-ASA treatment effectively inhibited UC-induced TNF-α, IL-1β, and IL-6 expressions in colon tissue. This present result shows that simultaneous treatment of treadmill exercise and 5-ASA potently suppressed inflammation in UC rats to a greater extent than either treatment alone. Patients with UC have been treated with a high dose of 5-ASA in combination with steroids to achieve potent anti-inflammatory effects [19]. However, due to many side effects, the combined use of steroids and high doses of 5-ASA is limited. In the current study, simultaneous treatment with 5-ASA and exercise was safe enough to limit steroid use. TNF-α is known to induce collagen accumulation and proliferation in intestinal myofibroblasts and is therefore associated with intestinal fibrosis [20]. In this study, mucosal damage caused by DSS administration was effectively prevented by combined treatment with treadmill exercise and 5-ASA. Collagen deposition in colonic tissue was also reduced. The current results show that the simultaneous treatment of treadmill exercise and 5-ASA ameliorated mucosal injury and reduced collagen deposition in UC mice.

The inflammatory response is a physio-immunological response to cell and tissue damage and is a process regulated through the MAPK multistep signaling pathway [21]. MAPKs regulate many cellular responses to pro-inflammatory cytokines or external stress signals. DSS activates the MAPK cascade, which controls the course of a wide range of cellular responses [22,23]. There are many forms of evidence that NF-κB is regulated by MAPKs [6,24]. The inducible transcription factor NF-κB, which plays a central role in inflammatory processes, is known to play a pivotal role in the pathogenesis of UC and Crohn’s disease [25,26]. Degradation of IκB-α protein by signal induction activates NF-κB, and this activated NF-κB complex moves to the nucleus from the cytoplasm, producing TNF-α, IL-1β, and IL-6 transcription factors [27,28]. This experiment showed that DSS caused phosphorylation of MAPK families such as JNK, ERK, and p38 in ulcerated colon tissue. Ultimately, MAPK cascade phosphorylation enhances activation of IκB-α, leading to NF-κB activation.

Results of this experiment, simultaneous treatment of treadmill exercise and 5-ASA, reduced the expression of activated MAPK cascades to a greater extent than treadmill exercise or 5-ASA treatment alone. Therefore, the combination treatment inhibited IκB-α and NF-κB activation in colon tissue. Physical exercise inhibits inflammation-activated NF-κB/MAPK signaling [29,30,31]. Likewise, 5-ASA, a major treatment for IBD, inhibited NF-κB and MAPK in IBD patients [32]. These results indicate that treadmill exercise and 5-ASA modulate the NF-κB/MAPK signaling pathway.

Apoptosis is known to be increased in mice with UC [33]. Excessive apoptosis causes mucosal epithelial damage, ulceration, and inflammatory cell infiltration [34]. Therefore, if apoptotic cells outnumber newly formed cells, UC healing may be delayed. Among the factors that determine whether a cell will progress to apoptosis or survive, the ratio of Bax to Bcl-2 acts is the most important factor [35]. The TUNEL assay detects apoptotic cell death and DNA fragmentation. Caspase activation is another important feature of apoptosis. Caspase-3 and caspase-9 are the most widely studied caspases and are the main executors of apoptosis. After UC onset, an imbalance between mitochondrial Bcl-2 and Bax results in enhanced cytochrome c release, causing caspase-3 and caspase-9 activation [36,37].

It is known that inhibition of NF-κB/MAPK cascade signaling inhibits the translocation of caspase-3, caspase-9, and Bax and consequently inhibits the release of cytochrome c from mitochondria [38,39]. The present study demonstrated that treadmill exercise and 5-ASA treatment inhibited the UC-induced increase in the Bax to Bcl-2 ratio, DNA fragmentation, and expression of cleaved caspase-3 and caspase-9 in colon tissue. These results show that treadmill exercise and 5-ASA treatment inhibit apoptosis in UC. Additionally, simultaneous treatment with treadmill exercise and 5-ASA inhibited apoptotic cell death much more strongly in UC mice than single treatment mice. These results show that treadmill exercise and 5-ASA effectively inhibit NF-κB/MAPK signaling activation, thereby suppressing apoptotic cell death.

## 4. Materials and Methods

### 4.1. Experimental Animals

Adult male C57BL/6N mice (*n* = 50) weighing 27 ± 3 g (10 weeks old) were bought from the Orient Bio Co. (Seongnam, Korea). These experimental animals were allowed free access to food and water and housed under the controlled temperature (23 ± 2 °C) and lighting (07:00–19:00) conditions. Mice were divided into one of the five groups (*n* = 10 per group): control group, UC induction group, UC induction and treadmill exercise group, UC induction and 5-ASA treatment group, and UC induction and treadmill exercise with 5-ASA treatment group. The entire process of this experiment was conducted in compliance with the animal care guidelines of the National Institutes of Health and the Korean Academy of Medical Sciences. The experimental design was approved by the Institutional Animal Care and Use Committee of Kyung Hee University and received the following approval number: KHUASP(SE)-17-159.

### 4.2. UC Induction

UC was induced using dextran sulfate sodium (DSS) as the following method [40]. DSS is a synthetic sulphated polysaccharide with anti-coagulant activity widely used to induce colitis. Except for the control group, mice consumed 2% DSS (MFCD00081551, MP Bio, Santa Ana, CA, USA) dissolved in water for 7 days.

### 4.3. Exercise Protocol and 5-ASA Treatment

After DSS administration, the mice in the exercise groups were forced to run on a motorized treadmill (EXER-6M treadmill, Columbus Instruments, Columbus, OH, USA) for 60 min once a day for 14 days. The exercise intensity was at an incline of 0° at a speed of 5 m/min for the first 10 min and at a speed of 8 m/min for the last 50 min. These settings represent low-intensity treadmill exercise. Mice in the drug administration groups were enema-injected once a day with 5-ASA (Asacol^®^, Daewoong Pharmaceutical, Seoul, Republic of Korea) using a polyethylene catheter with a capacity of 200 μL for 14 days. The 5-ASA concentrations in this study were referenced to those used in clinical use [41]. Considering this clinical dose and previous research results, a dose of 200 mg/kg was used [42].

### 4.4. Valuation of DAI

The extent of UC was assessed comprehensively using daily DAI and previously published scoring systems [43]. DAI was measured every 3 days from days 0 to 21 by monitoring weight change in body weight, consistency of stool, and total rectal bleeding.

### 4.5. Tissue Preparation

Mice were sacrificed 21 days after the first administration of DSS. Mice were anesthetized by injection of Zoletil 50^®^ (06516, 10 mg/kg, Vibac Laboratories, Carros, France) intraperitoneally. Laparotomy was performed, and distal colon segments from the same defined area were collected. To analyze colon length and morphological changes, segments of the colon were photographed and the colon was weighted. The collected colon tissues were immediately frozen at −80 °C, and some were used for Western blotting. The remaining colon tissues were used for histological staining after fixation with 4% paraformaldehyde solution in 100 mM phosphate buffer (pH 7.4) for 24 h. Colon tissues were dehydrated in various ethanol concentrations, embedded in paraffin, and sectioned at 5 μm in thickness by a paraffin microtome (Thermo Fisher Scientific, Somerset, NJ, USA). Colon tissue sections were then mounted on gelatin-coated slides and dried overnight in an oven at 37 °C.

### 4.6. H&E Staining

H&E staining was conducted for the evaluation of histological changes in colon tissue according to the method described earlier [44,45]. Paraffin slides were deparaffinized and rehydrated in xylene and ethanol. These slides were treated in Meyer’s hematoxylin (S3309, DAKO, Glostrup, Denmark) for 30 s and washed with water. These slides were incubated with eosin (318906, Sigma-Aldrich, St. Louis, MO, USA) for 10 s and washed with water. These slides were air-dried at room temperature and then sequentially treated in ethanol and xylene. Coverslips were covered onto the slides and mounted by Permount^®^ (SP15-100, Thermo Fisher Scientific, Waltham, MA, USA). Ulcerative colonic damage was evaluated with the Wallace microscopic score (Table 1) [46,47]. 

### 4.7. Masson’s Trichrome Staining

Masson’s trichrome staining for the detection of collagen fibers in colon tissue was done according to the method described earlier [48]. Paraffin slides were deparaffinized and rehydrated with xylene and ethanol. Following washing with water, these sections were treated with Bouin’s solution (HT10132, Sigma-Aldrich) at 56 °C for 1 h and incubated with Weigert iron hematoxylin (HT1079, Sigma-Aldrich) for 10 min. After washing with water, these sections were treated with Biebrich Scarlet-acid fuchsin (HT151, Sigma-Aldrich) and next incubated in phosphomolybdic-phosphotungstic acid solution (HT153, Sigma-Aldrich) for 10–15 min. These sections were incubated with aniline blue solution (B8563, Sigma-Aldrich) for 5 min and next with light-green solution for 1 min. After washing, these sections were incubated in glacial acetic acid solution for 5 min, then dehydrated quickly in 95% ethanol and then washed in xylene. Coverslips were covered onto the slides and mounted by Permount^®^ (Thermo Fisher Scientific). The fibrosis analysis was conducted according to the previously described method [49].

### 4.8. TUNEL Assay

TUNEL staining was done by the In Situ Cell Death Detection Kit^®^ (A23210, Roche, Mannheim, Germany) for the visualization of cell death in colonic tissue, according to the method described earlier [50]. Paraffin slides containing colonic sections were deparaffinized and rehydrated by xylene and ethanol. These sections were washed and then post-fixed using ethanol and acetic acid (2:1) solution. These sections were then treated by proteinase K (100 μg/mL), treated in 3% H_2_O_2_, permeabilized with 0.5% Triton X-100, and treated with TUNEL reaction mixture. After washing, these sections were visualized by Converter-POD with 0.03% 3,3′-diaminobenzidine. After counterstaining with Mayer hematoxylin (DAKO), these sections were mounted on gelatin-coated slides, air-dried overnight at room temperature, and next coverslipped by Permount^®^ (Thermo Fisher Scientific).

### 4.9. Immunohistochemistry

Immunohistochemistry for cleaved caspase-3 and caspase-9 was done according to the method described earlier [51]. Paraffin slides containing colonic sections were deparaffinized by xylene and ethanol and rehydrated. These sections were washed, and after boiling 10 mM citric acid (pH 6.0) for 10 min, these sections denatured. These sections were treated with mouse anti-cleaved caspase-3 (9661S) and anti-cleaved caspase-9 antibodies (9507S, Cell Signaling Technology Inc., Danvers, MA, USA) diluted 1:200 overnight. These sections were treated with biotinylated anti-rabbit (BA-1000) and anti-mouse secondary antibodies (BA-2000, Vector Laboratories, Burlingame, CA, USA) for 1 h. These sections were then treated with avidin-biotin-peroxidase complex (PK4010, Vector Laboratories) at room temperature for 1 h. By treating these sections with the solution containing 0.05% 3,3-diaminobenzidine and 0.01% H_2_O_2_ in 50 mM Tris-buffer (pH 7.6) for approximately 3 min, immunoreactivity was visualized. After counterstaining with Mayer hematoxylin (DAKO), these sections were mounted on gelatin-coated slides, air-dried overnight at room temperature, and coverslipped by Permount^®^ (Thermo Fisher Scientific).

### 4.10. Western Blotting

Western blotting was done according to the method described earlier [51]. After homogenizing with a lysis buffer, the colon tissues were centrifuged at 14,000 rpm for 20 min. After measuring protein content using a colorimetric protein assay kit (5000001, Bio-Rad, Hercules, CA, USA), protein was separated on SDS-polyacrylamide gels and transferred onto a nitrocellulose membrane. Mouse β-actin antibody (SC-47778, 1:1000; Santa Cruz Biotechnology, Santa Cruz, CA, USA), mouse TNF-α antibody (SC52746, 1:1000; Santa Cruz Biotechnology), mouse IL-1β antibody (SC52012, 1:1000; Santa Cruz Biotechnology), rabbit IL-6 antibody (ab208113, 1:2000; Abcam, Cambridge, UK), rabbit ERK 1/2 antibody (SC-514302, 1:1000; Santa Cruz Biotechnology), phosphorylated (p)-ERK 1/2 antibody (9101S, 1:2000; Cell Signaling Technology), mouse JNK antibody (SC-7345, 1:1000; Santa Cruz Biotechnology), mouse p-JNK antibody (SC-6254, 1:1000; Santa Cruz Biotechnology), rabbit p38 antibody (9212S, 1:2000; Cell Signaling Technology), rabbit p-p38 antibody (4511S, 1:2000; Cell Signaling Technology), rabbit Bax antibody (2772S, 1:2000; Cell Signaling Technology), rabbit Bcl-2 antibody (3498S, 1:2000; Cell Signaling Technology), rabbit NF-κB antibody (ab16502, 1:2000; Abcam), rabbit IκB-α antibody (GTX50468, 1:2000; GeneTex Corporation, CA USA), and rabbit p-IκB-α antibody (GTX50209, 1:2000; GeneTex) were treated on the membranes. The membranes were treated with horseradish peroxidase-conjugated secondary mouse antibody (PI2000, 1:5000) and rabbit antibody (PI1000, 1:1000) at room temperature for 1 h. An enhanced chemiluminescence detection system (#1705061, Santa Cruz Biotechnology) was used to visualize the bands.

### 4.11. Data Analysis

Histological observation was performed, and the percentages of TUNEL-positive, cleaved caspase-3-positive, and cleaved caspase-9-positive cells in each colonic tissue slice were measured by an Image-Pro^®^ plus computer-assisted image analysis system (Media Cybernetics Inc., Silver Spring, MD, USA) with a light microscope (Olympus BX51, Tokyo, Japan). To identify TUNEL-positive, cleaved caspase-3-positive, and cleaved caspase-9-positive cells, five visual fields in the colonic mucosa were randomly selected from each sample. At least 100 cells per field were observed under 200× magnification. The percentages of TUNEL-positive, cleaved caspase-3-positive, and cleaved caspase-9-positive cells were calculated using the following formula: positive cells/total cells × 100 (%). For the comparison of the relative expressions of proteins, the detected bands were quantified by a computer-assisted Image-Pro^®^ Plus analysis system (Media Cybernetics, Inc., Waltham, MA, USA). For the quantification, the value for the control group was set to 1.00. 

Statistical analysis was performed by one-way analysis of variance (ANOVA) followed by Duncan’s post-hoc test using IBM SPSS Statistics software (ver. 23.0, IBM Co., Armonk, NY, USA). Significance was set at *p* < 0.05, and all experimental results were expressed as the mean ± standard error of the mean. 

## 5. Conclusions

In conclusion, simultaneous treatment with treadmill exercise and 5-ASA improved histological damage, increasing body weight, colon weight, and colon length, and decreasing the DAI score and collagen deposition. Simultaneous treatment with treadmill exercise and 5-ASA inhibits pro-inflammatory cytokines and apoptosis after UC. It has been suggested that the beneficial therapeutic effects of treadmill exercise and 5-ASA on ulcerative bowel injury can be due to inhibition of NF-κB/MAPK signaling activation. Based on this study, the simultaneous treatment of treadmill exercise and 5-ASA can be considered as a new effective therapeutic approach for the management of UC.

## Figures and Tables

**Figure 1 ijms-25-05076-f001:**
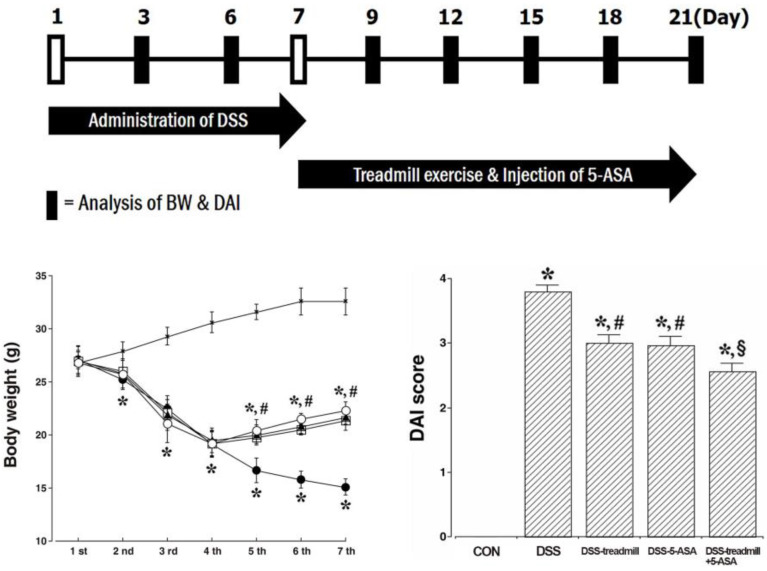
Changes in body weight and DAI score in UC mice. Upper panel: Experimental schedule. Left lower panel: Alteration of body weight during the experiment. (×) control group, (●) UC induction group, (□) UC induction and treadmill exercise group, (▲) UC induction and 5-ASA treatment group, (○) UC induction and treadmill exercise with 5-ASA treatment group. Right lower panel: Alteration of DAI score by DSS administration. (CON) control group, (DSS) UC induction group, (DSS-treadmill) UC induction and treadmill exercise group, (DSS-5-ASA) UC induction and 5-ASA treatment group, (DSS-treadmill+5-ASA) UC induction and treadmill exercise with 5-ASA treatment group. * means *p* < 0.05 when compared to the control group. # means *p* < 0.05 when compared to the UC induction group. § means *p* < 0.05 when compared to the UC induction and treadmill exercise group. DAI, disease activity index; UC, ulcerative colitis; 5-ASA, 5-aminosalicylic acid; DSS, dextran sulfate sodium.

**Figure 2 ijms-25-05076-f002:**
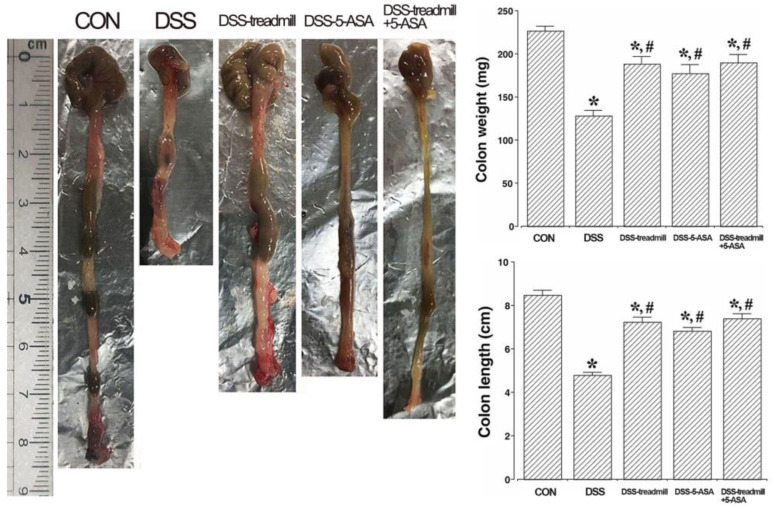
Changes in colon weight and length in UC mice. Left panel: Representative images of colon tissue. Right upper panel: Alteration of colon weight. Right lower panel: Alteration of colon length. (CON) control group, (DSS) UC induction group, (DSS-treadmill) UC induction and treadmill exercise group, (DSS-5-ASA) UC induction and 5-ASA treatment group, (DSS-treadmill+5-ASA) UC induction and treadmill exercise with 5-ASA treatment group. * means *p* < 0.05 when compared to the control group. # means *p* < 0.05 when compared to the UC induction group. UC, ulcerative colitis; 5-ASA, 5-aminosalicylic acid.

**Figure 3 ijms-25-05076-f003:**
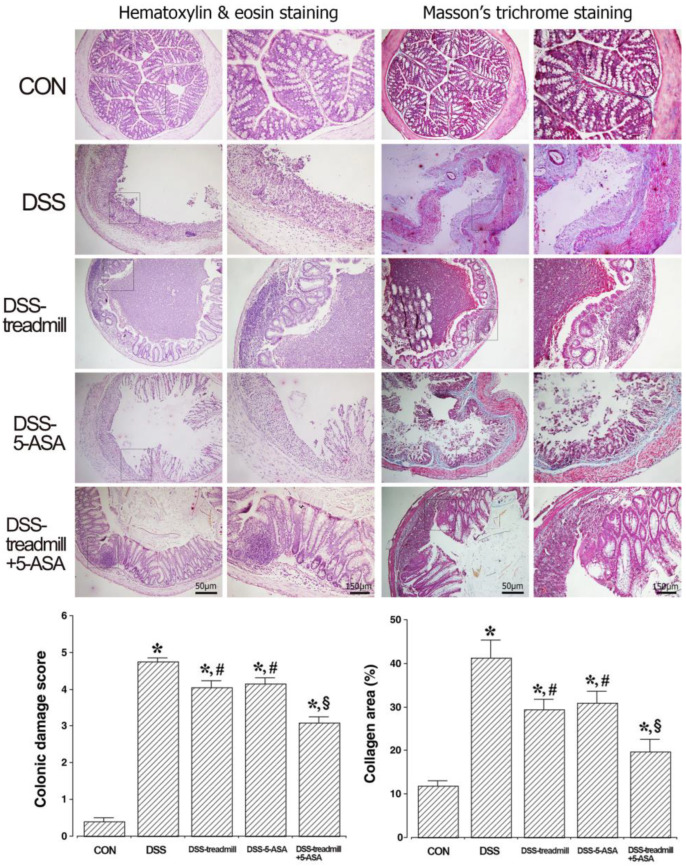
Changes in histological evaluation and collagen deposition in ulcerative colitis mice. Left panel: The sections were stained for hematoxylin (blue: myocytic nuclei) and eosin (pink: muscle fiber). Right panel: The sections were stained for Masson’s trichrome (pink: cytoplasm in muscle) and (blue: collagen fibers). (□) Observation range. The scale bars represent 50 μm and 150 μm. Lower column: colonic damage score, and collagen area in each group. (CON) control group, (DSS) UC induction group, (DSS-treadmill) UC induction and treadmill exercise group, (DSS-5-ASA) UC induction and 5-ASA treatment group, (DSS-treadmill+5-ASA) UC induction and treadmill exercise with 5-ASA treatment group. * means *p* < 0.05 when compared to the control group. # means *p* < 0.05 when compared to the UC induction group. § means *p* < 0.05 when compared to the UC induction and treadmill exercise group. UC, ulcerative colitis; 5-ASA, 5-aminosalicylic acid.

**Figure 4 ijms-25-05076-f004:**
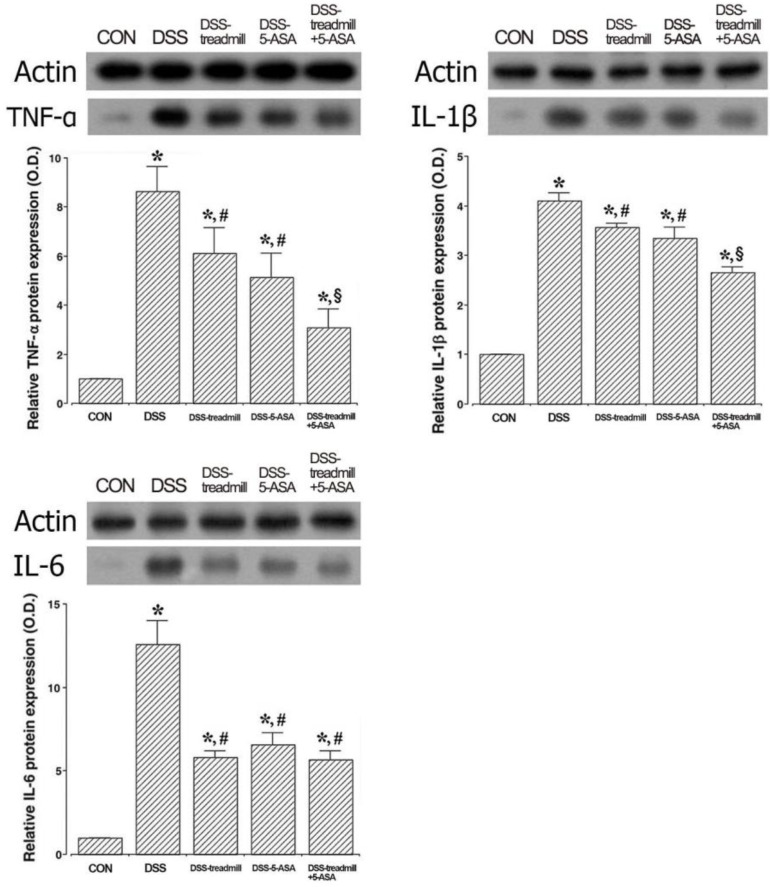
Changes in TNF-α, IL-1β, and IL-6 in UC mice. Left upper panel: Relative TNF-α expression. Right upper panel: Relative IL-1β expression. Left lower panel: Relative IL-6 expression. Actin was used as an internal control (46 kDa). (CON) control group, (DSS) UC induction group, (DSS-treadmill) UC induction and treadmill exercise group, (DSS-5-ASA) UC induction and 5-ASA treatment group, (DSS-treadmill+5-ASA) UC induction and treadmill exercise with 5-ASA treatment group. * means *p* < 0.05 when compared to the control group. # means *p* < 0.05 when compared to the UC induction group. § means *p* < 0.05 when compared to the UC induction and treadmill exercise group. TNF, tumor necrosis factor; IL, interleukin; UC, ulcerative colitis; 5-ASA, 5-aminosalicylic acid.

**Figure 5 ijms-25-05076-f005:**
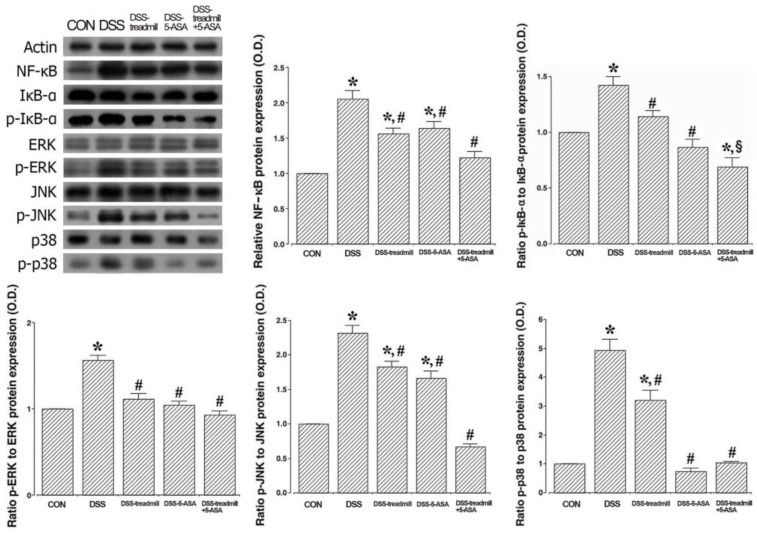
Changes in NF-κB, IκB-α expressions, and MAPK cascade including p38, JNK, and ERK in UC mice. Left upper panel: Representative expressions of NF-κB, IκB-α, MAPK cascade proteins. Actin was used as an internal control (46 kDa). Right upper panel: Relative expression of NF-κB and relative ratio of p-IκB-α to IκB-α protein expression. Lower panel: The relative ratio of p-ERK to ERK, p-JNK to JNK, and p-p-38 to p-38. (CON) control group, (DSS) UC induction group, (DSS-treadmill) UC induction and treadmill exercise group, (DSS-5-ASA) UC induction and 5-ASA treatment group, (DSS-treadmill+5-ASA) UC induction and treadmill exercise with 5-ASA treatment group. * means *p* < 0.05 when compared to the control group. # means *p* < 0.05 when compared to the UC induction group. § means *p* < 0.05 when compared to the UC induction and treadmill exercise group. NF-κB, nuclear factor-κB; IκB-α, NF-κB inhibitor-α; UC, ulcerative colitis; p, phosphorylated; 5-ASA, 5-aminosalicylic acid; MAPK, mitogen-activated protein kinase; JNK, c-Jun N-terminal kinase; ERK, extracellular signal-regulated kinase.

**Figure 6 ijms-25-05076-f006:**
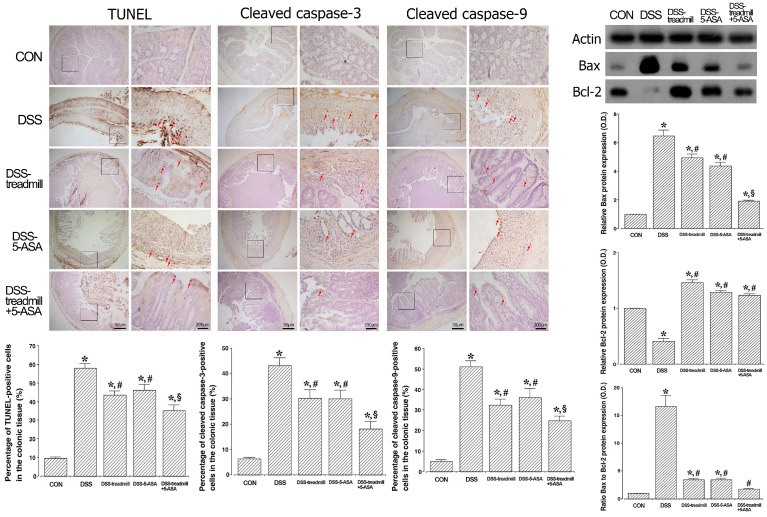
Changes in apoptotic cell death in UC mice. Left upper panel: Photomicrographs of TUENL-positive cells. Middle upper panel: Photomicrographs of cleaved caspase-3,-9-positive cells. Red arrow represents each item-positive cell. Right upper panel: The relative protein ratio of Bax to Bcl-2. Actin was used as an internal control (46 kDa). Lower panel: Percentage of TUNEL-positive and cleaved caspase-3,-9-positive cells. (CON) control group, (DSS) UC induction group, (DSS-treadmill) UC induction and treadmill exercise group, (DSS-5-ASA) UC induction and 5-ASA treatment group, (DSS-treadmill+5-ASA) UC induction and treadmill exercise with 5-ASA treatment group. * means *p* < 0.05 when compared to the control group. # means *p* < 0.05 when compared to the UC induction group. § means *p* < 0.05 when compared to the UC induction and treadmill exercise group. TUNEL, terminal deoxynucleotidyl transferase-mediated deoxyuridine triphosphate nick end labeling; Bax, Bcl-2-associated X protein; Bcl-2, B-cell lymphoma-2; UC, ulcerative colitis; 5-ASA, 5-aminosalicylic acid.

**Table 1 ijms-25-05076-t001:** Wallace histological colonic damage score.

Score	Appearance
0	Normal
1	Damage limited to surface epithelium
2	Focal ulceration limited to mucosa
3	Focal, transmural inflammation and ulceration
4	Extensive transmural ulceration and inflammation bordered by normal mucosa
5	Extensive transmural ulceration and inflammation involving entire section

## Data Availability

All data generated in this study are included in this manuscript.

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
