# Peer review of "Simultaneous Treatment of 5-Aminosalicylic Acid and Treadmill Exercise More Effectively Improves Ulcerative Colitis in Mice"

_ijms, 2024, doi:10.3390/ijms25105076_

Round 1

Reviewer 1 Report

Comments and Suggestions for Authors

Title: Simultaneous treatment of treadmill exercise and 5-aminosalicylic acid more effectively improves dextran sulfate sodium-mediated ulcerative colitis in mice

The authors have tried to address the impact of exercise on DSS-induced colitis. However, significant concerns are highlighted in the major and minor comment sections.  

Major comments:

·         In the method section, it is mentioned that 5-ASA is administered rectally; what was the rationale for administering rectally when it can be administered with oral gavage? Please substantiate the advantages and disadvantages of oral gavage over rectal administration.

·          In the method section it is mentioned that the 5-ASA concentrations in this study used is comparable to clinical use [40]. Considering this clinical dose and previous research results, a dose of 200mg/kg was used [41]. According to reference 41, 5-ASA is administered through the oral route. Therefore, rectal administration should have a solid scientific justification.

·         Histological evaluation and collagen deposition histology pictures should be quantitative. This is crucial for a comprehensive understanding of the results and to ensure the validity of the conclusions. Therefore, an evaluation of colon inflammation and crypt distortion scores must be quantified and represented with clear statistical significance. Is there any quantitative measurement score available for collagen deposition? If so, it must be supported by previously published scientific articles.

·         Figure 4's statistical analysis clearly shows that the DSS+5-ASA+Tread mill group did not show any significance compared to the DSS+tread mill group with regard to TNF-alpha and IL-6 protein expression. However, as shown in the result section, two of three major proinflammatory cytokine expressions were not influenced by treadmill + 5-ASA intervention. Therefore, the effect is of minimum significance.

·         Reference [29] has shown a direct effect of Irisin on Nf-kB and MAPK when animals were treated with irisin in LPS-induced lung injury. Therefore, using this reference to support reduction in these signaling events in exercise is wrong. Moreover, there is no detailed discussion on how exercise may mediate inhibiting these signaling pathways, with thorough literature support missing.

·         The conclusions derived from this study appear to be overstretched and warrant reassessment. For a more comprehensive understanding of the implications of the study findings, it is essential to conduct statistical analyses that compare all groups rather than selecting groups based on convenience. This approach will provide a more thorough insight into the impact of the study results.

Minor comments:

·         The title should read: Simultaneous treatment of treadmill exercise and 5-aminosalicylic acid……improves dextran sodium sulfate-mediated ulcerative colitis. It is not dextran sodium sulfate…Please correct the title.

·         In Figure 1, a graphical representation depicting the experimental protocol should not show an injection of DSS. Rather, it should show the administration of DSS. DSS is not injected into the body to induce ulcerative colitis; it is administered through drinking water.

·         All the bar diagrams must be labeled clearly with all the groups, e.g., 1) control, 2) DSS, 3) DSS + treadmill, 4) DSS + 5-ASA, 5) DSS + treadmill + 5-ASA.

·         Provide the details of the treadmill exercise regimen and explain the type of exercise the approach may belong to.

·         Give the details (including the version) of the statistical software used in this study to carry out the analysis

Author Response

We sincerely appreciate for your kind advice and comments to our manuscript. We revised the manuscript according to the reviewer’s comments. We added new experimental data, and modifications were expressed in red. Please check the attached file for all modifications.

Reviewer 2 Report

Comments and Suggestions for Authors

General comments:

In this manuscript, the authors reported that simultaneous treatment of 5-aminosalicylic acid and treadmill exercise more effectively improves DSS-induced ulcerative colitis in mice. The study design and result presentation are reasonable. The topic is interesting, and the data provided useful information in this field. The article contains innovative elements, it is well written and in my opinion after small corrections it can be published in the IJMS.

Special comments:

Title: should be changed. (Simultaneous treatment of 5-aminosalicylic acid and treadmill exercise more effectively… sounds more meaningful)

Results:

·      Line 81: top panel? In the figure changes in the body weight are presented in the lower left panel)

·      Line 91: bottom panel (add right)

·      Figure 5: enlarge stars and hashtags on the figure because they are illegible.

·      Figure 6: The figure is too small, and it is not possible to read what is on the charts, enlarge the font and preferably the whole figure.

Discussion:

·      Line 316: remove excessive spaces

·      Line 339: move the conclusions at the end of the article as point 5 of the conclusion

Materials and Methods

·      DSS: explain the abbreviation

·      Add reagents catalogue numbers

·      Remove excessive breaks between subsections

·      If you add references, it would be necessary to write as described earlier and not next (subsections 4.6-4.11)

Informed consent statement: add “Not applicable”. 

Author Response

(The authors gave the same response as above.)

Round 2

Reviewer 1 Report

Comments and Suggestions for Authors

Thanks for providing the detailed rebuttal. The manuscript's readability and scientific merit have increased.